# Association between General Anesthesia and Root Canal Treatment Outcomes in Patients with Mental Disability: A Retrospective Cohort Study

**DOI:** 10.3390/jpm12020213

**Published:** 2022-02-03

**Authors:** Guan-Yu Chen, Zhi-Fu Wu, Yi-Ting Lin, Kuang-I Cheng, Yu-Ting Huang, Shun-Te Huang, Arief Hargono, Chung-Yi Li

**Affiliations:** 1Department and Graduate, Institute of Public Health, College of Medicine, National Cheng Kung University, Tainan 70101, Taiwan; kindtaco@gmail.com; 2Department of Anesthesiology, Kaohsiung Medical University Hospital, Kaohsiung Medical University, Kaohsiung 807, Taiwan; aneswu@gmail.com (Z.-F.W.); Kuaich@gmail.com (K.-I.C.); 3Department of Anesthesiology, College of Medicine, Kaohsiung Medical University, Kaohsiung 807, Taiwan; 4Department of Dentistry, Division of Special Care Dentistry, Kaohsiung Medical University Hospital, Kaohsiung 807, Taiwan; melody061@yahoo.com.tw (Y.-T.L.); shuntehuang@gmail.com (S.-T.H.); 5School of Dentistry, Kaohsiung Medical University, Kaohsiung 807, Taiwan; 6Department of Medical Research, Division of Medical Statistics and Bioinformatics, Kaohsiung Medical University Hospital, Kaohsiung Medical University, Kaohsiung 807, Taiwan; stakmuh@gmail.com; 7Department of Epidemiology, Faculty of Public Health, Universitas Airlangga, Surabaya 60115, Indonesia; arief.hargono@gmail.com; 8Department of Public Health, College of Public Health, China Medical University, Taichung 40402, Taiwan; 9Department of Healthcare Administration, College of Medical and Health Science, Asia University, Taichung 41354, Taiwan

**Keywords:** dental care outcome, disability, general anesthesia, root canal treatment

## Abstract

In the population of individuals with a disability, mental illness patients can be uncooperative during dental treatment; thus, general anesthesia has been widely applied during dental procedures. This study aims to investigate the association between general anesthesia and the outcomes of root canal treatment in patients with disability. Teeth treatment records of patients with disability from Kaohsiung Medical University Hospital Research Database and electronic database from January 2005 to December 2018 were used in this retrospective cohort study. The authors conducted analysis comparing root canal treatment outcomes under general anesthesia and non-general anesthesia, indicated by endodontic re-treatment or post-treatment teeth extraction. Over the 9-year follow-up period, root canal treatment outcomes representing a cumulative survival rate of 87.68% and 74.51% in the general anesthesia group and non-general anesthesia group, respectively, were found. After adjustment for potential confounders, the teeth with general anesthesia showed a substantially and significantly reduced HR of root canal treatment failure at 0.24 (95% confidence interval, 0.12 to 0.49). Our study supported the notion that root canal treatment with general anesthesia may entail substantial reduction of treatment failure in patients with disability.

## 1. Introduction

Approximately 15% of the world’s population live with some form of disability, of whom 2–4% experience significant difficulties in functioning according to the World Health Organization’s 2011 report on disability [1]. The affected population is at a higher risk of oral health problems and poorer oral hygiene compared to those without disabilities [2,3]. Moreover, the disabled patients usually suffered from impaired physiological or psychological conditions, which leads to uncooperative behavior and emotional swings during oral treatment, and additional techniques have been indicated for provision of dental services to disabled individuals. General anesthesia (GA) is one of the applicable techniques to facilitate dental procedures in patients with dental fear or challenging behavior [4]. There have been reports which found improvements of oral health-related quality of life in patients with disabilities after dental treatment under GA [5,6,7]. Meanwhile, the demand for dental treatment for special-needs patients under GA continues to increase [8]. Over time, the provision of GA for special-needs patients has changed from dental clinics to general hospitals [8], and the volume of ambulatory anesthesia services for pediatric dentistry has also increased [9]. In addition, the mortality rate of dental treatment under GA decreased gradually [10], and more and more practitioners consider it to be a safe procedure with adequate devices and trained personnel. 

Despite the high acceptance of GA in providing dental treatment to patients with disability, there was limited evidence concerning the potential benefit of administering GA during dental treatment. The reasons for limited evidence from previous studies included legitimate difficulty in conducting clinical trials, unavailability of sufficient number of disabled participants, inadequate period of follow up, and involvement of various confounding factors in observational studies. Although some research reported that dental treatment with GA in patients with special needs could result in a successful rate similar to that of the general population [11,12], these studies were only descriptive and adopted no control group [11,12]. We therefore conducted this retrospective observational cohort study with improved methodology to provide better evidence for the potential benefit of GA in root canal treatment in patients with disability.

## 2. Materials and Methods

We performed this retrospective observational cohort study of root canal treatment (RCT) failure in association with GA in patients with disability and special needs in dental treatment. Approval of this study was obtained from the Institutional Review Board at Kaohsiung Medical University Chung-Ho Memorial Hospital (approval numbers: KMUHIRB-E(I)-20200008). Design and conduct of this study have followed the Strengthening the Reporting of Observational Studies in Epidemiology (STROBE) guideline.

### 2.1. Sources of Data

Data were derived from Kaohsiung Medical University Hospital Research Database (KMUHRD) and merged with the Electronic Medical Records (EMR) of Kaohsiung Medical University Chung-Ho Memorial Hospital, which consists of electronic medical records from one medical center hospital within KMU health system, established in 1957 and with a predominance in southern Taiwan. The KMUHRD provides a comprehensive database with coverage on ambulatory care, hospital admissions, dental services, drug-dispensing records, and laboratory patient data. The EMR of Kaohsiung Medical University Chung-Ho Memorial Hospital contains the information of registry of disability, with severity of disability being classified by the International Classification of Function, Disability, and Health [13].

### 2.2. Study Sample

In this study, we enrolled patients with various types of disability (i.e., intellectual disability, dementia, autism, and chronic mental illness) who received RCT at outpatient settings of Kaohsiung Medical University Chung-Ho Memorial Hospital from January 2005 to December 2018. To confirm the accuracy of “first-visit” RCT records, we used the data from January 2005 to December 2009 to double check that the study population had no previous RCT records. The follow-up period of this study was designed to be at least one year. Therefore, the enrolled period was from January 2010 to December 2017. We also excluded the RCT records of deciduous teeth to prevent misjudgment of failure events and survival period of the teeth.

### 2.3. Exposure

Information on the main exposure, namely GA, was obtained from the medical claims. The medical claim of GA was based on the breath systems by open, semi-open, semi-close, and close system, under intravenous infusion or inhalation of general anesthetic agents (Appendix A). The GA group comprised patients who received RCT and GA on the same date. The patients who received RCT but without GA were considered as the non-GA group in this study.

### 2.4. Outcome Measure

The primary outcome was the failure of RCT, which was defined as either endodontic re-treatment or extraction after first-ever RCT. Identification of endodontic re-treatment and extraction was based on the medical orders of Taiwan’s NHI program (Appendix A). 

### 2.5. Covariates

This study collected the covariates concerning both patients and teeth. Patients’ characteristics included age, sex, type and severity of disability, and comorbidity including periodontitis, hypertension, dyslipidemia, diabetes mellitus, and ischemic heart disease. Meanwhile, several studies had reported that age, periodontitis, hypertension, dyslipidemia, diabetes mellitus, and ischemic heart disease exhibited increased failure risk of RCT [14,15,16]. The age was defined by the day of RCT. Information of comorbidity was based on the International Classification of Diseases, Ninth Revision, Clinical Modification (ICD-9-CM) codes retrieved from the inpatient and outpatient claim data in the 1-year period prior to the index RCT (Appendix A). Information of type and severity of disability was obtained from the EMR of the hospital according to the “official certificate of disability” approved by the Taiwan government. The “official certificate of disability” was granted based on the People with Disabilities Rights Protection Act, after processes of evaluation and assessment by the committee composed of professionals from medicine, social work, special education and employment counseling and evaluation, and classification of the type and severity of disability with ICD-9-CM and International Classification of Function, Disability, and Health [13]. The information of the tooth position was determined according to the two-digit FDI (French: Fédération Dentaire Internationale) world dental federation notation that is also known as ISO-3950 Notation [17], and was categorized into incisor and canine, premolar, and molar, and distinguished upper and lower teeth.

### 2.6. Statistical Analysis 

The study results were presented with continuous variables as mean ± SD, and with categorical variables as number (percentage). The characteristics of patient between GA and non-GA groups were compared by Student’s *t*-test and Chi-square test for continuous and categorical variables, respectively.

We calculated the cumulative survival rate using Kaplan-Meier methods of RCT teeth without failure event over up to 9 years. We also constructed and compared survival curves between GA and non-GA groups. A log-rank test was then performed to test the difference between the survival curves. Cox proportional hazards models that account for cluster data were used to account for intercorrelation of teeth from the same patient [18] and to estimate the hazard ratios (HRs) and corresponding 95% confidence interval (CI) of RCT failure. In the calculation of adjusted Cox hazards models, based on literature review and biological plausibility, we selected covariates as potential risk factors of the association between the cumulative success rate and GA. This study also adjusted the difference between the two groups by adding the type and severity of the disability into the estimation of adjusted Cox proportional model. The follow-up period was set from the date of first-ever RCT to the occurrence of RCT failure (i.e., either endodontic re-treatment or extraction) or censoring, which was defined as the last day of 2018. We also performed Cox proportional hazards models treating endodontic re-treatment and extraction as the study endpoints separately. In doing so, we used Cox proportional hazards model with Fine and Gray’s method to account for “extraction” as a potential competing risk event in the analysis [19].

The data analysis was performed with either SAS (version 9.4; SAS institute, Gary, NC, USA) or R software version 4.1.0 (R Foundation for Statistical Computing, Vienna, Austria). We used the “survival” package for the calculation of cumulative survival rate with Kaplan-Meier method, as well as for the Cox regression model with cluster data. We also used the “ggsurvplot” function from the “survminer” package to generate the Kaplan-Meier plot. The threshold of statistical significance was set at two-tailed *p* value less than 0.05.

## 3. Results

### 3.1. Participants

Participants included a total of 60,014 outpatient claims of RCT for patients with disability at Kaohsiung Medical University Chung-Ho Memorial Hospital between January 2005 and December 2018. We retained 21,873 first-visit RCT records of permanent teeth from January 2010 to December 2017 based on a washout period from January 2005 to December 2009; and kept at least one year for follow up after RCT. This study finally enrolled 280 teeth from 108 patients who received GA for RCT, and 217 teeth from 106 patients without GA from RCT. The flow chart of sample enrollment is shown in Figure 1.

### 3.2. The Baseline Characteristics Comparison between the Two Groups

The baseline characteristics of the study patients and teeth are listed in Table 1. Patients in the GA group were younger and suffered lesser comorbidity, and there was a slight male dominance. Most (≥70%) patients had a disability type of intellectual disability. Patients of GA group had higher prevalence of intellectual disability and autism, but lower prevalence of dementia and chronic mental illness. They also suffered from higher prevalence of disability of severe and extremely severe forms. The distributions of position of teeth treated with RCT were similar between the two groups. Patients in the GA group experienced much lower incidence of endodontic re-treatment events (2.5% GA group vs. 11.1% non-GA group), extraction events (3.2% GA group vs. 10.6% non-GA group), and overall RCT failure (5.4% GA group vs. 21.7% non-GA group).

### 3.3. The Cumulative Survival Rate of Teeth between Two Groups

The Kaplan–Meier survival curves demonstrated a 9-yr RCT cumulative survival rate of 87.7% (95% CI, 77.3 to 99.4) and 74.5% (95% CI, 68.1 to 81.5) for the GA group and non-GA group, respectively (Table 2). Log-rank test indicated a significant difference in survival curves between the two groups (*p* < 0.0001) (Figure 2). The Kaplan-Meier survival curves for endodontic re-treatment and extraction as separate endpoints are presented in Appendix A. The results similarly showed significantly higher survival rate of endodontic re-treatment and extraction individually in the GA group.

We also performed the sub-stratification analysis of the disability patients. The analysis of Kaplan–Meier survival curves and log-rank test revealed no difference of RCT failure according to severity and type of the disability. Furthermore, we performed sub-stratification analysis according to the type of disability. The significant difference in cumulative survival rates between GA and non-GA groups was observed only for intellectually disabled patients, but not for patients with autism or chronic mental illness, although GA also tended to show a superior outcome. The non-significant difference in the two cumulative survival curves was likely due to limited numbers of study patients in these two groups.

### 3.4. The Risk Factors Analysis for Cumulative Survival Rate of Teeth 

Compared to the non-GA group, the GA group was associated with a significantly reduced risk of RCT failure in both crude and covariate adjusted analyses with an HR 0.24 (95% CI, 0.13 to 0.44) and 0.24 (95% CI, 0.12 to 0.49), respectively (Table 3). The analytical results for endodontic re-treatment and extraction as separate endpoints are presented in Appendix A, which also showed significantly lower risk of failure associated with GA, with an HR of 0.16 and 0.38, respectively.

## 4. Discussion

To the best of our knowledge, this is the first study to evaluate the association between GA administration and outcome of RCT in patients with disability who usually need special aids in dental treatment. The GA group was found to be associated with a better survival rate of teeth with RCT, and such seemingly beneficial outcome is independent of known risk factors for RCT failure. Our study showed that the cumulative five-year survival rate of RCT with general anesthesia was high at 94.4%, which revealed an even better result compared to the figures reported in the population-based cohort study from 1997 to 2010 in Taiwan (a five-year cumulative survival rate of 89.3%) [20]. 

Although most dentists believe that GA for non-cooperative patients may result in better treatment outcomes, limited evidence concerning such speculation has been available from the previous literature. In 2014, Cousson et al. reported the outcome of endodontic treatment of 225 permanent teeth under GA in the special needs unit of the University Dental Hospital of Clermont-Ferrand. It revealed that 87% of the teeth had a successful outcome after a follow-up period of 1 month to 2 years [11]. In 2017, Chang et al. also reported favorable outcomes of single visit endodontic and restoration treatments under GA for 381 teeth in 203 special needs patients [12]. However, these two studies lacked control groups, which makes interpretation difficult. Our study not only used the non-GA control group in making the comparison but also employed a larger sample size and longer follow-up period to detect endpoints of interest. 

Based on the Donabedian’s structure–process–outcome conceptual framework [21], the higher quality of dental treatment for special needs patients under GA could be the potential etiology of better outcome. The process of RCT comprises sequential steps of preparing the area, accessing and cleaning the roots, and shaping and filling the canals, which all require that the patient cooperate with dentist in order to achieve a high quality of debridement and isolation for cleaning and disinfection [22]. Compared to the patients under GA, the disability patients without GA could be intolerant to the stimulations during the dental treatment. Therefore, there could be a potential residual infection in the root canal or inadequate isolation during the process of dental treatment, which could ultimately lead to failure of RCT [23,24].

Although GA for dental treatment is thought to be a safe procedure nowadays [9], the potential risk involved in sedation should not be overlooked. Very few mortalities and permanent neurologic damages associated with GA were reported in previous studies, which were largely due to inappropriately monitoring patients, administration or dose error of the anesthetic agents, or inadequate environment [25,26]. Additionally, the occurrence of intraoperative apnea resulting in hypoxia is common for dental treatment under sedation, which could potentially result in severe complications [26]. After discontinuing the administration of anesthetic agents, post-discharge excessive somnolence, nausea, and emesis were considered common postoperative adverse events [27]; however, deaths are rarely reported after discharging [28]. There was a decreasing trend of mortality associated with dental sedation under the clinical guidelines with adequate environment, facilities, and qualified practitioners [10]. In our study, no severe complications were noted.

We included several comorbidities that might potentially affect RCT failure in the multivariate model, and none of these comorbidities were significantly associated with increased risk of RCT failure, which was different from other studies [15,29,30]. This could be the result of younger study population in our study compared to other studies [15,29,30], which could also contribute to lower prevalence rate of comorbidities in our study cohort. Furthermore, these other studies evaluated the risk factors of RCT outcome with large study populations of up to fifty thousand [30,31], while our study population was comparatively small. Some studies reported that available scientific evidence remains inconclusive as to whether diabetes and/or cardiovascular disease(s) may be associated with endodontic outcomes [14]. However, although our data reveal risk potential of included comorbidities, they showed no significant association with the outcome of RCT. 

Despite the improvement in methodology and the novelty of the findings of our study, several limitations involved in our study need to be noted. First, a potential confounding by indication might be involved in the selection of patients for GA. There was no specific criteria in the selection of general anesthesia. Generally, the patients whose disability is very severe are likely to receive GA in RCT, and these severely disabled patients are likely to suffer from worse outcomes after RCT, mainly due to the difficulties in maintaining satisfactory oral health experienced by these patients [32]. However, the potential for confounding by indication may have resulted in an underestimation rather than overestimation of the potential beneficial effect of general anesthesia. Moreover, we have adjusted the type and severity in our analysis, which further reduced the potential for confounding by indication. Second, because we used EMR in our analysis without having an interview with each of the study patients, there could be a potential for loss of follow-up among our study patients who might have endodontic re-treatment or extraction at other medical institutions. However, individuals with a disability tend to have a high level of continued care through the medical services and have empathetic, compassionate, and responsible professionals at the initial visit [33]. Therefore, the effectiveness of care delivery in our study cohort significantly reduces the chance of loss to follow up. Third, the information of histories of comorbidities might be incomplete in our EMR. The data of comorbidities were derived from the outpatient department of one single medical center, and underreporting of comorbidity is possible for study patients who made clinical visits to other medical institutions. Because the patients included in our study are young, the prevalence of comorbidity is considered low, which minimized the potential information bias. Furthermore, there were no available records in our database concerning side effects such as nausea or vomiting in post-operation. Fourth, our study might be subject to residual confounding due to incomplete adjustment for all known prognostic factors for RCT. Previous studies noted a number of clinical characteristics, including preoperative signs and symptoms, lesions of periodontal involvement, adequate root-filing length, and material of root-end filling, which are associated with prognosis of RCT [15,34]. Furthermore, the individual and environment factors might also affect outcomes of RCT, including non-parental caregiver, cooperation level, and soft diet [12]. Although the above factors were left unadjusted in our analysis, the potential residual confounding by these factors should not be sizeable, because their associations with the use of general anesthesia should be small.

## 5. Conclusions

In conclusion, our study revealed that administration of GA to the patients with disability was significantly associated with better survival of their teeth with RCT. Further studies should be conducted to further explore the mechanism with which GA may yield better RCT outcomes in patients with disability. 

## Figures and Tables

**Figure 1 jpm-12-00213-f001:**
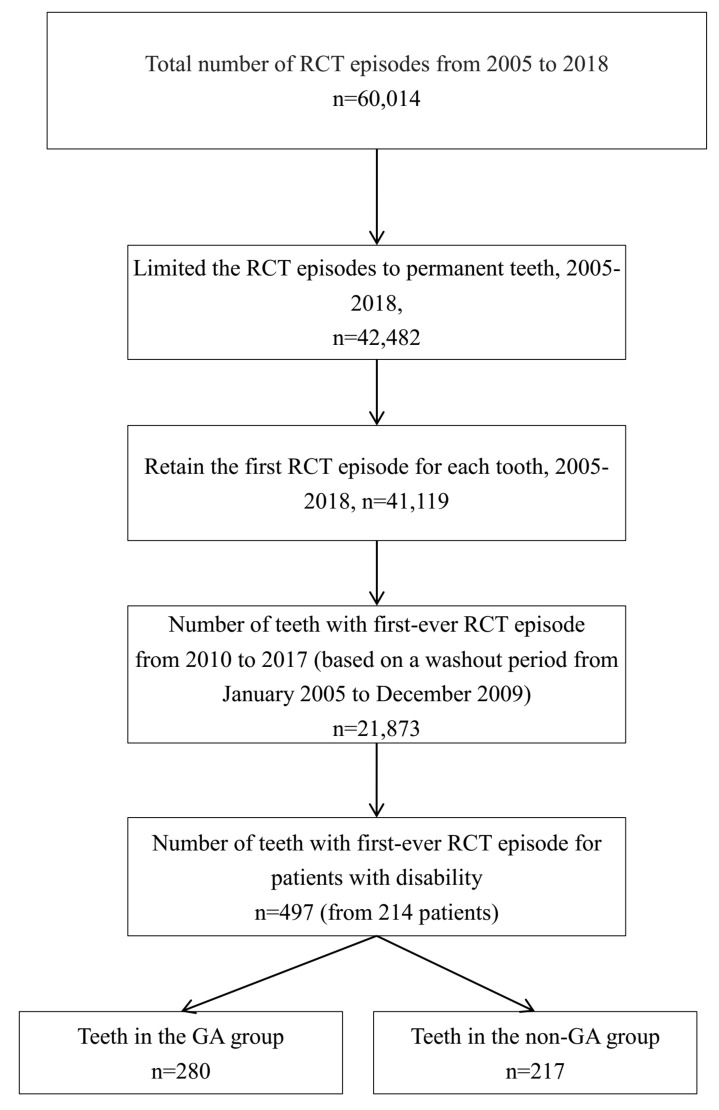
Diagram showing the selection of study sample population. GA, general anesthesia; RCT, root canal treatment.

**Figure 2 jpm-12-00213-f002:**
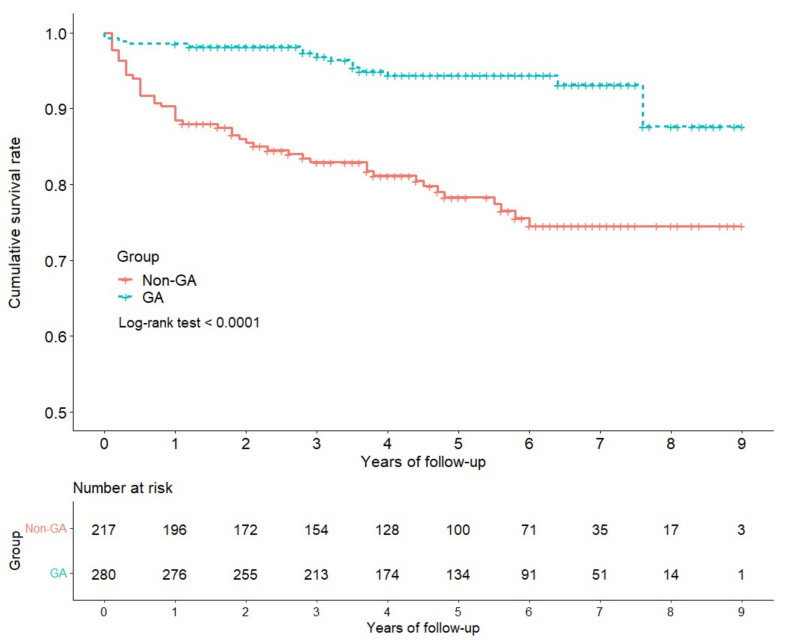
Comparison of 9-year cumulative survival rate of teeth receiving root canal treatment with and without general anesthesia. GA, general anesthesia; non-GA, not general anesthesia.

**Table 1 jpm-12-00213-t001:** Comparisons of characteristics between RCT patients with and without general anesthesia.

Characteristics	GA (*n* = 280)	Non-GA (*n* = 217)	*p*-Value
Patients number	108	106	
Age	25.3 ± 13.7	31.5 ± 16.5	<0.001 **
Gender (male)	158 (56.4)	117(53.9)	0.577
Comorbidity			
Periodontitis	110 (39.3)	133(61.3)	<0.001 **
Hypertension	0 (0.0)	6(2.8)	0.005 **
Dyslipidemia	3 (1.1)	11(5.1)	0.008 **
Diabetes mellitus	3 (1.1)	11(5.1)	0.008 **
Ischemia heart disease	0 (<0.1)	1(0.5)	0.256
Disability type			<0.001 **
Mental retard	210 (75.0)	152 (70.0)	
Dementia	2 (0.7)	18 (8.3)	
Autism	45 (16.1)	21 (9.7)	
Chronic mental illness	23 (8.2)	26 (12.0)	
Severity of disability			0.030 *
Mild	16 (5.7)	24 (11.1)	
Moderate	86 (30.7)	77 (35.5)	
Severe	108 (38.6)	79 (36.4)	
Extremely severe	70 (25.0)	37 (17.1)	
Teeth position			0.367
Incisor and canine	101 (36.1)	82 (37.8)	
Premolar	75 (26.8)	67 (30.9)	
Molar	104 (37.1)	68 (31.3)	
Maxillary teeth (upper)	153 (54.6)	114 (52.5)	0.640
Endpoints			
Endodontic re-treatment	7 (2.5)	24 (11.1)	<0.001 **
Extraction	9 (3.2)	24 (11.1)	<0.001 **
Failure of RCT	15 (5.4)	47 (21.7)	<0.001 **

Data showed as mean ± SD or *n* (%). * *p* < 0.05. ** *p* < 0.01.

**Table 2 jpm-12-00213-t002:** Cumulative survival rate of RCT failure in patients treated with and without general anesthesia according to various endpoints.

Follow-Up (Years)	Definition of Failure					
	Endodontic Re-Treatment		Extraction		Failure of RCT	
	GA	Non-GA	GA	Non-GA	GA	Non-GA
	%	%	%	%	%	%
1	99.64	92.63	98.93	95.85	98.57	88.48
2	99.64	91.18	98.57	93.85	98.21	85.54
3	98.76	89.57	98.11	92.83	96.88	82.93
4	97.21	89.57	97.15	91.08	94.38	81.19
5	97.21	88.08	97.15	89.70	94.38	78.30
6	97.21	88.08	96.30	85.99	94.38	74.51
7	95.98	88.08	96.30	85.99	93.16	74.51
8	95.98	88.08	91.23	85.99	87.68	74.51
9	95.98	88.08	91.23	85.99	87.68	74.51
*p* for Log-rank test	0.0001		0.0012		<0.0001	

Failure of RCT = event of either endodontic re-treatment or extraction.

**Table 3 jpm-12-00213-t003:** Hazard ratio of failure of RCT in association with general anesthesia and other characteristics of patients or teeth.

Variables	NFRCT	CHR (95%CI)	*p* Value	AHR (95%CI)	*p* Value
General anesthesia					
No	47	Reference group		Reference group	
Yes	15	0.24 (0.13–0.44)	<0.001**	0.24 (0.12–0.49)	<0.001 **
Age		1.00 (0.98–1–02)	0.982	0.97 (0.93–1.00)	0.071
Periodontitis					
No	27	Reference group		Reference group	
Yes	35	1.48 (0.84–2.62)	0.177	1.31 (0.71–2.41)	0.381
Hypertension					
No	61	Reference group		Reference group	
Yes	1	1.38 (0.15–12.69)	0.776	0.82 (0.06–10.90)	0.885
Dyslipidemia					
No	57	Reference group		Reference group	
Yes	5	3.28 (1.64–6.54)	0.001**	1.89 (0.59–6.10)	0.287
Diabetes mellitus					
No	58	Reference group		Reference group	
Yes	4	2.46 (0.97–6.27)	0.059	0.80 (0.22–2.88)	0.734
Disability type					
Mental retard	49	Reference group		Reference group	
Dementia	4	1.92 (0.74–4.98)	0.178	4.15 (0.73–23.61)	0.108
Autism	4	0.45 (0.17–1.21)	0.114	0.39 (0.14–1.08)	0.071
Chronic mental illness	5	0.69 (0.22–2.20)	0.535	0.82 (0.27–2.49)	0.720
Severity					
Mild	6	Reference group		Reference group	
Moderate	22	0.91 (0.32–2.61)	0.865	0.99 (0.32–3.11)	0.992
Severe	26	0.94 (0.33–2.68)	0.910	1.06 (0.35–3.22)	0.918
Extreme severe	8	0.53 (0.17–1.63)	0.266	0.65 (0.19–2.23)	0.489

NFRCT, number of teeth with failure event of root canal treatment; CHR, crude hazard ratios; AHR, adjusted hazard ratios; GA, general anesthesia; N/A, not applicable; Failure of RCT = event of either endodontic re-treatment or extraction. * *p* < 0.05. ** *p* < 0.01.

## Data Availability

Not applicable.

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
