# Peer review of "Association between General Anesthesia and Root Canal Treatment Outcomes in Patients with Mental Disability: A Retrospective Cohort Study"

_jpm, 2022, doi:10.3390/jpm12020213_

Round 1

Reviewer 1 Report

The title - Association between general anesthesia and root canal treatment outcomes in patients with disability, a retrospective cohort study is not clear.

The authors should state de definition for disability patients and patients with other invalidating pathologies, psychiatric disorders. It is not clear for the readers the characteristics of the included subjects.

Which were the criterias considering the necesity of general anesthesia for the included subjects (physical disability#psychiatric disorder)?

Were any of the diagnosed cases of disability in danger for the anesthetic substances?

Was there any sub-stratification of the disability patients in the two groups?

There is no information regarding the regarding the potential complication after RCT assessment.

there were some literatures revealed no correlation between some systemic disease (i.e., diabetes mellitus, cardiovascular disease) and the out- 263 come of endodontic treatment [14,32,33] - I do not agree with the following statement, the authors should revise the literature.

The conclusion of the study is vague. Also, the authors state that more than 20000 subjects are not enough in order to answer the study s purpose.

Author Response

Dear reviewer:

I am very grateful to your comments for the manuscript. According with your advice, we amended the relevant part in manuscript. Some of your questions were answered below.

Reviewer1.

  1. The authors should state de definition for disability patients and patients with other invalidating pathologies, psychiatric disorders. It is not clear for the readers the characteristics of the included subjects.

Ans:

Thanks for your suggestion. I added illustration in the section "2.4 covariates" as following (page 3, line 119 to 126): Information of type and severity of disability was obtained from the EMR of the hospital according to the " official certificate of disability” approved by the Taiwan government. The “official certificate of disability” was granted based on People with Disabilities Rights Protection Act, after processes of evaluation & assessment by the committee composed of professionals from medicine, social work, special education and employment counseling and evaluation, and classify the type and severity of disability with ICD-9-CM and International Classification of Function, Disability and Health [13].

  1. Which were the criterias considering the necesity of general anesthesia for the included subjects (physical disability#psychiatric disorder)?

Ans:

There was no specific criteria in the selection of general anesthesia. We illustrated this limitation in discussion as following. And we would add the following sentence to make the description more specific: " There was no specific criteria in the selection of general anesthesia ". (page 9, line 280)  

  1. Were any of the diagnosed cases of disability in danger for the anesthetic substances?

Ans:

We added the following sentence at the discussion (page9, line:266) "There were no severe complications were noted in the database ".  

  1. Was there any sub-stratification of the disability patients in the two groups?

Ans:

Yes, we did the sub-stratification analysis of the disability patients as following. And we add illustration at "section 3.3". (page 6, line 194 to 201) First, we analyze the difference of RCT failure according to severity, and type of the disability. The results were shown in the following: (you could see the figures in the profile: answer to Reviewer 1. docx)

Further, we performed sub-stratification analysis according to type of disability. The significant difference in cumulative survival rates between GA and non-GA groups was observed only for intelligent disability patients, but not for patients with autism or chronic mental illness, despite GA also tended to show a superior outcome. The non-significant difference in the two cumulative survival curves was likely due to limited numbers of study patients in these two groups.

  1. There is no information regarding the regarding the potential complication after RCT assessment.

Ans:

There were no available records in our database concerning side effects such as nausea or vomiting in post-operation. We add this illustration at the third limitation in discussion (page 10, line 299-300).   

6. there were some literatures revealed no correlation between some systemic disease (i.e., diabetes mellitus, cardiovascular disease) and the out- 263 come of endodontic treatment [14,32,33] - I do not agree with the following statement, the authors should revise the literature.

Ans:

The statement could be too strong, and I cited the wrong reference 32, and 33. I have changed the statement into " some literatures reported that available scientific evidence remains inconclusive as to whether diabetes and/or cardiovascular disease(s) may be associated with endodontic outcomes." (page 9, line 274-276).  

7. The conclusion of the study is vague. Also, the authors state that more than 20000 subjects are not enough in order to answer the study s purpose.

Ans:

We revised the conclusion as the following to address the Reviewer’s concern. Besides, we agree with the Reviewer that our sample is quite enough, it may not be necessary to replicate our study findings in more studies with even larger study sample sizes.

“In conclusion, our study revealed that administration of GA to the patients with disability was significantly associated with better survival of their teeth with RCT. Further studies should be conducted to further explore the mechanism with which GA may yield better RCT outcomes in patients with disability.”

Reviewer 2 Report

Thank you for giving me the opportunity to review your article.

  1. It needs full form for RCT in text.  There was no full form for RCT, only abbreviation. 
  2. The author used the word 'disability' in the title and text. However, the subjects of study were patients with mental disabilities. It doesn't seem to include patients with physical disabilities. I think that 'mental disability' is better than 'disability' in the title. 
  3. In this study, the number of cases was more than the number of patients. It means that duplicated patients were involved and follow-up periods could overlap. Such patients may have more dental service and it can affect the results. 
  4. This study divided groups based on anesthesia cases. Could the same patient be included in another group according to the anesthesia case in patients with receiving several RCT? 
  5. It is also better to see if similar results are obtained in patients who have received only one treatment.

Author Response

Dear reviewer:

I am very grateful to your comments for the manuscript. According with your advice, we amended the relevant part in manuscript. Some of your questions were answered below.

Reviewer 2. 

  1. It needs full form for RCT in text.  There was no full form for RCT, only abbreviation.

Ans:

    We have included the full name of RCT in the revised manuscript (page 2, line 73). 

  1. The author used the word 'disability' in the title and text. However, the subjects of study were patients with mental disabilities. It doesn't seem to include patients with physical disabilities. I think that 'mental disability' is better than 'disability' in the title.

Ans:

    Thanks for your suggestion. We have followed the Reviewer’s comment by revising our title as “Association between general anesthesia and root canal treatment outcomes in patients with mental disability: A retrospective cohort study”

  1. In this study, the number of cases was more than the number of patients. It means that duplicated patients were involved and follow-up periods could overlap. Such patients may have more dental service and it can affect the results.

Ans:

 The unit of analysis in the present study was “teeth” rather than “patient with mental disability”. It is true that some patients included in our study received multiple RCTs for their teeth, and these patients could have poor oral health, which may have led poor outcome of the RCTs. To account for the potential inter-correlation among teeth from the same patient, we have adopted appropriate statistical methods to account for such data inter-correlation.   

  1. This study divided groups based on anesthesia cases. Could the same patient be included in another group according to the anesthesia case in patients with receiving several RCT?

Ans:

     Yes, for example, one disability patient could have three teeth received RCTs, one of the tooth could be performed RCT without general anesthesia, while the other two tooth received RCTs under general anesthesia at other appointed time. In our study, 15 (dedicated 56 teeth) of the total 214 patients in our study database meted this situation.  We have adopted appropriate statistical methods to account for such data inter-correlation.

  1. It is also better to see if similar results are obtained in patients who have received only one treatment.

Ans:

The unit of analysis in the present study was “teeth” rather than “patient with mental disability”. If only one tooth was used for a patient, we are afraid that the sample size may not be adequate for the analysis. Because the Cox proportional hazards models that account for cluster data can account for the intercorrelation of teeth from the same patient, we tended to retain the current approach that allowed multiple teeth with RCT from the same patient being included in the analysis.

Round 2

Reviewer 1 Report

The manuscript has bee improved and can be considered for publication.

Reviewer 2 Report

Thank you.